# Multi-Scale Factor Image Super-Resolution Algorithm with Information Distillation Network

**Yu Cheng, Shuai Chen \*, Zeyu Liao and Niujun Zhou**

School of Automation, Nanjing University of Science and Technology, Nanjing 210094, China; cy22021159@njust.edu.cn (Y.C.); liao1758521090@gmail.com (Z.L.); 13770558290@sina.cn (N.Z.)
\* Correspondence: chenshuai@njust.edu.cn

**Abstract:** Deep convolutional neural networks with strong expressive ability have achieved impressive performances in single-image super-resolution algorithms. However, excessive convolutions usually consume high computational cost, which limits the application of super-resolution technology in low computing power devices. Besides, super-resolution of arbitrary scale factor has been ignored for a long time. Most previous researchers have trained a specific network model separately for each factor, and taken the super-resolution of several integer scale factors into consideration. In this paper, we put forward a multi-scale factor network (MFN), which dynamically predicts the weights of the upscale filter by taking the scale factor as input, and generates HR images with corresponding scale factors from the weights. This method is suitable for arbitrary scale factors (integer or non-integer). In addition, we use an information distillation structure to gradually extract multi-scale spatial features. Extensive experiments suggest that the proposed method performs favorably against the state-of-the-art SR algorithms in term of visual quality, PSNR/SSIM evaluation indicators, and model parameters.

**Keywords:** single image super-resolution; information distillation; multi-scale factor network

## 1. Introduction

In computer vision, single image super-resolution (SISR) is currently a hot research topic, which reconstructs a high-resolution (HR) image from a low-resolution (LR) image through image processing methods in the same scene [1]. SISR is widely used in the fields of medicine, transportation, and remote sensing. Since one LR image can generate several HR images, SISR has no unique solution [2]. To address this problem, numerous image SR methods based on deep neural network architectures have been proposed and have shown prominent performance.

Since deep learning shows strong advantages in various computer vision tasks, Dong et al. [3,4] achieved feature extraction, nonlinear matching, and image reconstruction by a three-layer network. VDSR [5] expanded dramatically the depth of the network to 20 by stacking multiple layers to enhance the receptive field. At the same time, Kim et al. [6] proposed DRCN for the first time to apply recursive learning to SR tasks. Tai et al. [7] first adopted a DRRN to reduce parameters. In addition, Tai et al. [8] used a persistent memory network (Mem-Net) that stacks with a densely connected structure to resolve the dependency problem. EDSR [9] removed the batch normalization (BN) layer and used the residual scaling to speed up the training. Zhang et al. [10] added densely connected blocks to the residual to form a residual dense network (RDN). The RDN makes full use of global and local features to enhance SR performance. GFSR [11] used a gradient-guided and multi-scale feature network for image super-resolution. HRFFN [12] designed an enhanced residual block (ERB) containing multiple mixed-attention blocks (MABs) to boost the representative ability of the network. The above algorithms all increased the network depth to upgrade the quality of images [13]. Kim Seonjae proposed two

lightweight neural networks with a hybrid residual and dense connection structure to improve the super-resolution performance [14]. However, they usually ignore the problems such as memory consumption and the network is prone to overfitting.

As for the upsampling methods, most use post-upsampling, and need to train a single model for each magnification. Dong et al. first upscaled the resolution as the output size in SRCNN [3,4]. Then they proposed FSRCNN [15], which used a transposed convolution at the end of the network to finish the upsampling operation. Afterwards, Lai et al. [16,17] believed that when the scale factor is large (×8), it is difficult to restore image texture through a one-step operation. So, they proposed Lap-SRN [16,17], which progressively extracted image features and achieved image super-resolution. Shi et al. [18] first used the sub-pixel convolution to upscale the size of feature map for reducing computation. In recent years, many methods have used sub-pixel convolution, such as EDSR [1] and RCAN [19]. However, these SISR methods only consider certain integer scale factors (×2, ×4, ×8). We need to train a module for each scale factor. LESRCNN [20] can obtain a high-quality image by a model for different scales. Few previous works have discussed how to implement super-resolution of the arbitrary scale factor. Meta-SR [21] first proposed to use a single model to achieve multiple magnification.

To solve the above problems, we propose a multi-factor image super-resolution network based on information distillation (IDMF-SR) to realize arbitrary scale SR with the smallest parameters. IDMF-SR mainly includes two parts: a feature learning block and a multi-scale factor upsampling block. The feature learning block is a collection of several information distillation modules. In the information distillation structure, four $3 \times 3$ convolutions are used to extract image features. After each convolutional layer, a channel split operation divides the extracted features into two parts, and one part is sent to the next convolutional layer, while another part of the feature is retained. We adopted a channel attention mechanism based on contrast-aware. Then the retained feature maps are fused through concatenation at the end. The feature fusion is carried out according to the importance of the feature maps. In the upsampling steps, we adopted a multi-factor network, which includes position projection, weight prediction, and feature mapping. As shown in Figure 1, our IDMF-SR achieves better visual results compared with state-of-the-art methods.

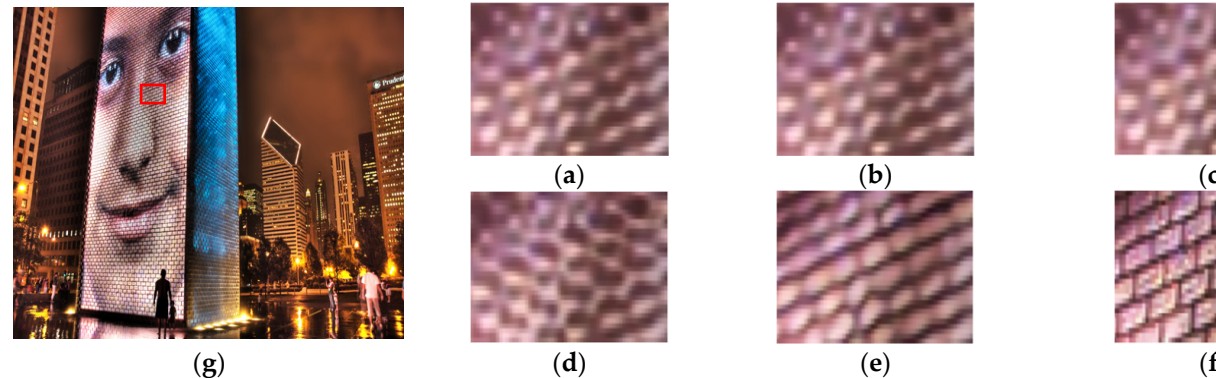

**Figure 1.** Visual results under ×4 upscale factor. (**a**) VDSR; (**b**) Lap-SRN; (**c**) Meta-SR; (**d**) RCAN; (**e**) IDMF-SR; (**f**) HR; (**g**) Urban100 img_76 (3×).

The contribution of this paper can be summarized as the following four points:

- We propose the multi-scale factor image super-resolution network (IDMF-SR) based on information distillation for significantly reducing the number of parameters. Our IDMF-SR is an end-to-end network model, which can utilize hierarchical features more than previous CNN-based methods and balance performance against applicability;
- We put forward a new information distillation network to gradually extract and cascade features. IDN divides the feature map extracted from each layer into two

parts. One of the parts flows into the next convolutional layer, and the retrained part is cascaded in the end;

- We propose a contrast-aware channel attention mechanism (CCAM) in the information distillation network. The traditional channel attention mechanism obtains the importance of the channel through the squeeze-and-excitation module, which is conducive to improving the PSNR value. Our CCAM can further enhance image details, such as edges, textures, and structures;
- IDMF-SR is inspired by meta-learning, and the network achieves image magnification by predicting filter weights by scale factors. Only training one network model can realize the image magnification at any multiple, which is conducive to application in the real scene.

## 2. Materials and Methods

### 2.1. Network Structure

IDMF-SR mainly includes two parts: a deep feature learning block and a multi-scale factor up-sampling block, as shown in Figure 2. First, a Conv-3 is used to extract coarse image features. The key component of IDMF-SR utilizes multiple-stacked information distillation blocks (IDBs). After each information distillation block, the feature maps flow into the next IDB and flows on to the last IDB. When several convolution operations are completed, the retained multi-scale feature maps are fused through concatenation. The upsampling module mainly includes position projection, weight prediction, and feature mapping, as shown in Figure 2. Details are introduced in Section 2.3.

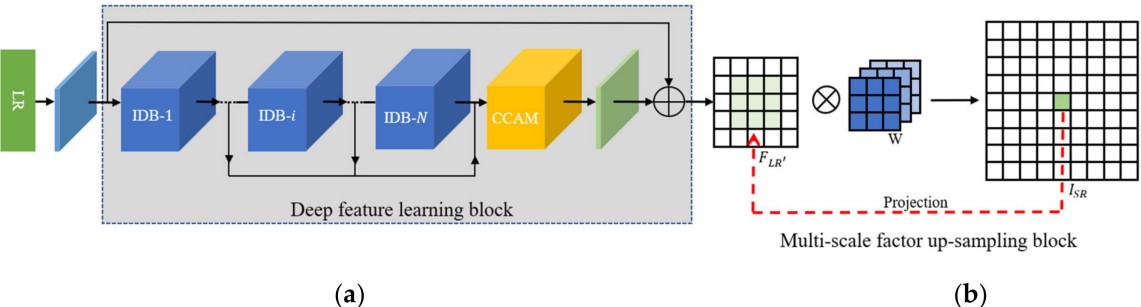

**Figure 2.** Network architecture of multi-factor image super-resolution based on information distillation (IDMF-SR). (**a**) The blue box represents Conv-3; (**b**) The green box represents Conv-1.

### 2.2. Information Distillation Module

In Figure 3, the information distillation block firstly uses four $3 \times 3$ convolutions to progressively extract image features. After each convolution, a channel split operation is used to divide the feature maps into two parts. One of the parts flows into the next convolutional layer, and the other part is retained. Finally, the retained feature maps are concatenated to flow into the next IDB. Assuming that the input of the $n\_th$ information distillation module is $F\_in$, the process can be expressed as Formulas (1)–(4).

$$F^n_{r\_1}, F^n_{c\_1} = Split^n_1\left(C^n_1\left(F^n_{in}\right)\right) \tag{1}$$

$$F^n_{r\_2}, F^n_{c\_2} = Split^n_2\left(C^n_2\left(F^n_{c\_1}\right)\right) \tag{2}$$

$$F^n_{r\_3}, F^n_{c\_3} = Split^n_3\left(C^n_3\left(F^n_{c\_2}\right)\right) \tag{3}$$

$$F^n_{r\_4} = C^n_4\left(F^n_{c\_3}\right)) \tag{4}$$

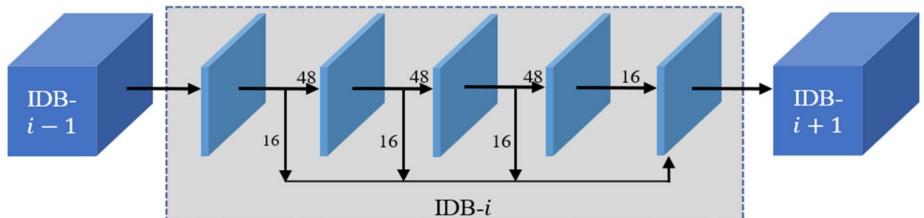

**Figure 3.** Information distillation module.

$C_1^n$ represents the first convolutional layer of the *n_th* information distillation module, $C_2^n, C_3^n, C_4^n$, and so on. $Split_1^n$ represents the first channel split layer of the *n_th* information distillation module. $F_{r\_1}^n$ represents the first retained feature maps, and $F_{c\_1}^n$ represents the first coarse feature, which is fed into the next calculation unit. After each level of convolutional layer, the feature maps are divided into two parts. Two-thirds flow into the next level, and one-third are retained. Table 1 shows the hyperparameter in the information distillation module. We set $3 \times 3$ as the kernel size in the convolutional layer. The output channels numbered 64, 48, and 16 are the convolutional layer. The number of the retained feature maps are 16, after four convolutional layers, the number of the output channels is also 64. The convolution kernel and stride follow the common operations in the SISR method.

**Table 1.** Convolutional parameter setting in the information distillation module.

| Layer | Number of Input_Channel | Kernel_Size | Stride | Number of Output_Channel |
|-------|------------------------|-------------|--------|--------------------------|
| C_1 | 64 | 3 | 1 | 64 |
| C_2 | 48 | 3 | 1 | 64 |
| C_3 | 48 | 3 | 1 | 64 |
| C_4 | 48 | 3 | 1 | 16 |

Next, we connect the previously retained feature maps $F_r^n$, which can be expressed by Formula (5):

$$F_{distilled}^n = Concat\left(F_{r_1}^n, F_{r_2}^n, F_{r_3}^n, F_{r\_4}^n\right) \tag{5}$$

We discard the traditional channel attention mechanism and add contrast variables to the original channel attention. In low-level image tasks, such as image super-resolution reconstruction, the contrast-based channel attention mechanism can enhance image details, such as edges and textures. In Figure 4, the contrast is the sum of the standard deviation and the mean. Assuming that the input feature has *C* feature maps, the size of each feature map is $H \times W$, and the input is expressed as $X = [x_1, x_2, \ldots x_c, \ldots x_C]$, and the contrast is calculated as Formula (6):

$$Z_c = H_{GC}(x_c) = \sqrt{\frac{1}{HW} \sum_{(i,j) \in x_c} \left(x_c^{i,j} - \frac{1}{HW} \sum_{(i,j) \in x_c} x_c^{i,j}\right)^2 + \frac{1}{HW} \sum_{(i,j) \in x_c} x_c^{i,j}} \tag{6}$$

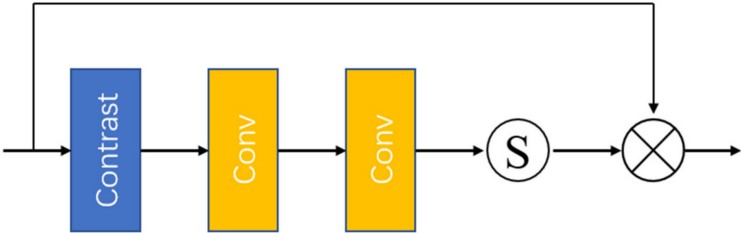

**Figure 4.** Contrast-based channel attention mechanism (S is sigmoid function).

Among them, $H_{GC}(\bullet)$ represents the global contrast information measurement function of the feature map, $\sqrt{\frac{1}{HW}\sum\limits_{(i,j)\in x_c}(x_c^{i,j}-\frac{1}{HW}\sum\limits_{(i,j)\in x_c}x_c^{i,j})^2}$ represents the standard deviation, and $\frac{1}{HW}\sum\limits_{(i,j)\in x_c}x_c^{i,j}$ represents the mean. IDMF-SR can effectively enhance image texture and improve SISR performance by using the contrast-based channel attention mechanism.

### 2.3. Multi-Factor Upsampling Module

The upsampling module mainly includes position projection, weight prediction, and feature mapping. The Location Projection projects pixels onto the LR image. The Weight Prediction Module predicts the weights of the filter for each pixel on the SR image. Finally, the Feature Mapping function maps the feature on the LR image with the predicted weights back to the SR image to calculate the value of the pixel. After $I_{LR}$ extracts image features through the information distillation module, the output feature map is $F_{LR'}$, and the network finally outputs $I_{SR}$. According to the principle that a pixel on the HR image can be back-projected to the $I_{LR}$, pixel $(i,j)$ on the $I_{SR}$ can be determined by a pixel $(i',j')$ on the LR image and the filter weight. Therefore, the upsampling module needs a specific filter to match $(i',j')$ and $(i,j)$. The formula is shown in Formula (7). $\Phi(\bullet)$ is the mapping function from $I_{LR}$ to $I_{HR}$. $F_{LR'}(i',j')$ represents the pixel on the $I_{LR}$, and $I_{SR}(i,j)$ represents the pixel on the $I_{SR}$.

$$I_{SR}(i,j) = \Phi\big(F_{LR'}(i',j'), W(i,j)\big) \tag{7}$$

(1) Position projection

Position projection is to back-project $I_{SR}$ onto $F_{LR'}$, as shown in Figure 5. The value of pixel $(i,j)$ on $I_{SR}$ is determined by the point $(i',j')$ on $F_{LR'}$.the relationship between these two pixels is expressed by Formula (8).

$$(i',j') = T(i,j) = \left(\left\lfloor\frac{i}{r},\frac{j}{r}\right\rfloor\right) \tag{8}$$

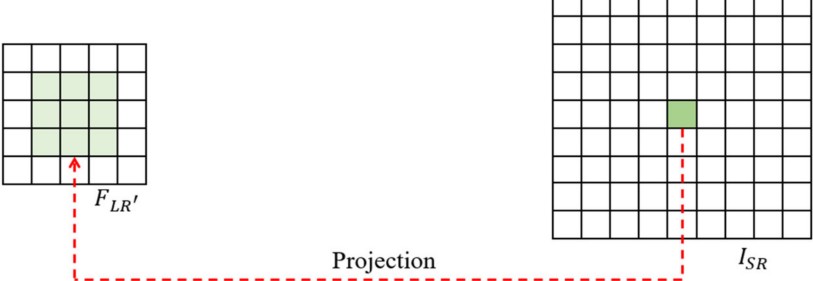

**Figure 5.** Location projection schematic diagram.

Among them, $T(\bullet)$ is the conversion function, which converts the point $(i,j)$ into $(i',j')$. $\left\lfloor\frac{i}{r},\frac{j}{r}\right\rfloor$ is floor function, and $r$ is scale-factor. It can be seen that adding a scale factor to calculate the relationship between two pixels is suitable for SISR with any scale factor.

The Location Projection can upscale the feature maps with arbitrary scale factor. The scale factor $r$ is divided into two types: integer and non-integer. When $r$ is an integer, for example, when $r$ is 2, one pixel in the LR image can determine two pixels in the HR image, as shown in Figure 6a. When the scale factor is a non-integer, for example, $r$ is 1.5, one pixel in the LR image determines one or two pixels in the HR image, as shown in Figure 6b. No matter whether $r$ is an integer or a non-integer, there is always a unique point on the LR image corresponding to a point on the SR image, and these two pixels are called the most relevant pixel pair.

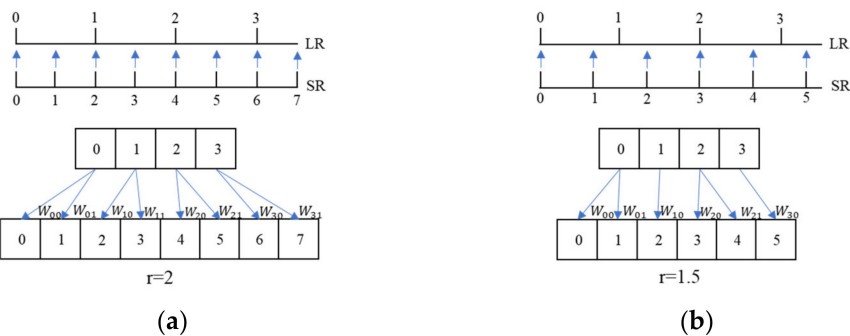

**Figure 6.** Pixel mapping schematic diagram. (**a**) r = 2; (**b**) r = 1.5.

Different from the typical upscale module, we use a network to predict the filter weights. This process is called weight prediction, expressed by Formula (9):

$$W(i,j) = \varphi\left(I_{i,j}; \theta\right) \tag{9}$$

$\varphi\left(I_{i,j}; \theta\right)$ represents the weight prediction process, $I_{i,j}$ is the input of the weight prediction network, $\theta$ is the parameter of the weight prediction network, and $W(i,j)$ is the weight at the pixel $(i,j)$. At the pixel $(i,j)$, the input $I_{ij}$ of $\varphi(\bullet)$ can be expanded to the relative offset of $(i',j')$, which is expressed as followed by Formula (10):

$$I_{ij} = \left( \frac{i}{r} - \left\lfloor \frac{i}{r} \right\rfloor, \frac{j}{r} - \left\lfloor \frac{j}{r} \right\rfloor \right) \tag{10}$$

To train multiple scale factors for a network, we add scale factor $r$ to the expression of $I_{ij}$. Assuming that the image is upscaled by 2 and 4, then $I_{SR_2}$ and $I_{SR_4}$ are obtained. Arbitrary pixels $(i,j)$ on $I_{SR_2}$ will have the same filter weights and position projection coordinates as $(2i, 2j)$ pixels on $I_{SR_4}$. Therefore, we improve the $I_{ij}$ expression to the Formula (11):

$$I_{ij} = \left( \frac{i}{r} - \left\lfloor \frac{i}{r} \right\rfloor, \frac{j}{r} - \left\lfloor \frac{j}{r} \right\rfloor, \frac{1}{r} \right) \tag{11}$$

The weight prediction network is the key of IDMF-SR. Its input is the vector $I_{ij}$ related to the pixel $(i,j)$, and the weight matrix is generated through several fully connected layers and activation layers, as shown in Figure 7. Finally, the size of the weight matrix is $(inC, outC, k, k)$, $inC$ represents the number of $F_{LR'}$, $outC$ represents the number of channels of the predicted HR image, and $k$ is the size of kernel.

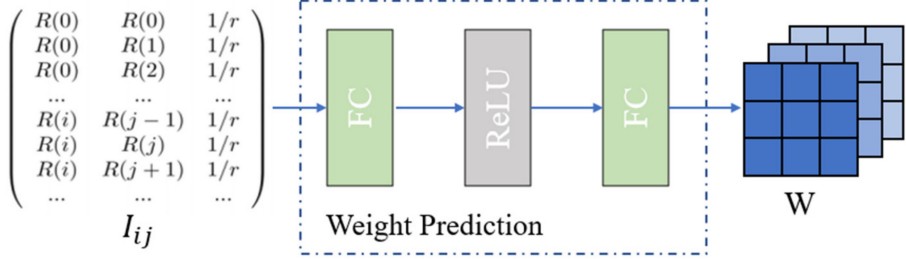

**Figure 7.** Weight prediction network schematic diagram.

(2)　Feature mapping

We got the feature of $(i', j')$ on the LR image from $F_{LR'}$. We predict the filter weights with weight prediction network. The last step is feature mapping, that is, $F_{LR'}$ is mapped

onto the SR image, as shown in Figure 8. We multiply $F_{LR'}(i', j')$ and the weights to get $\Phi(\bullet)$, as expressed in Formula (12):

$$\Phi\left(F^{LR}(i', j'), W(i, j)\right) = F_{LR'}(i', j') \bullet W(i, j) \tag{12}$$

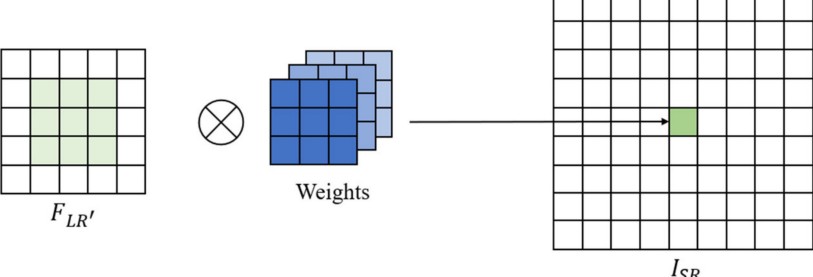

**Figure 8.** Feature mapping schematic diagram.

### 2.4. Datasets and Evaluation Metrics

In our experiments, we train the network by DIV2K [22], which contains 800 high-quality images. We use Set5 [23], Set14 [24], BSD100 [25], and Manga109 [26] for evaluation. There are two metrics to evaluate the performance of the SR, such as peak signal-to-noise ratio (PSNR) and structure similarity (SSIM) [27]. We calculate the values on the Y channel transformed from YCbCr space. As for the degradation methods, we use bicubic downsampling on the Matlab platform, the original HR image is downscaled to obtain the LR image. We randomly cropped into image patches with size 192 × 192, which are used as input for network training.

### 2.5. Implementation Details

In the experiment, we set the optimizer as the Adam, where $\beta_1 = 0.9$, $\beta_2 = 0.999$, and $\epsilon = 10^{-8}$. The initial learning rate is set to $2 \times 10^{-4}$, and the learning rate is reduced by half for every $2 \times 10^5$ steps. The loss function uses the $L_1$ and the kernel size is generally set to 3 × 3. The number of 3 × 3 convolutional layers of the information distillation module is set to 4. The IDMF-SR is implemented by the Pytorch framework. The code runs in the Windows 10 operating system, which is equipped with NVIDIA GeForce GTX1080Ti. We use CUDA9.0 and CuDNN7.1 to accelerate training.

## 3. Results

This section will analyze IDMF-SR from PSNR and SSIM evaluation indicators and visual effects.

### 3.1. Comparison of Objective Evaluation Indicators

In this experiment, SRCNN [3,4], VDSR [5], Lap-SRN [16,17], LESRCNN [20], and Meta-SR [21] are selected as reference methods for comparative experiments. BSD100 is selected as the test dataset, and the upscaling factor is 1.1–1.9. In Table 2, we compare the PSNR value between IDMF-SR and state-of-the-art SR methods. It can be seen that IDMF-SR is slightly better than the PSNR value of Meta-SR [21], but has a similar PSNR value to RCAN [19]. Compared with LESRCNN [20], IDMF-SR almost comprehensively outperforms LESRCNN. Under ×2, the performance is slightly different. It can be seen from the PSNR and SSIM that IDMF-SR has improved PSNR and SSIM performance indicators compared to Meta-SR [21] and RCAN [19] methods. As shown in Table 3, the PSNR index of IDMF-SR can reach 40.15 dB on the Manga109 test data set with factor of 2, which is 2.8 dB, 0.91 dB and 1.42 dB higher than Meta-SR [21], RCAN [19], and LESRCNN [20].

**Table 2.** The PSNR value of IDMF-SR under non-integer upscale factors.

| Method | ×1.1 PSNR | ×1.2 PSNR | ×1.3 PSNR | ×1.4 PSNR | ×1.5 PSNR | ×1.6 PSNR | ×1.7 PSNR | ×1.8 PSNR | ×1.9 PSNR |
|---|---|---|---|---|---|---|---|---|---|
| Bicubic | 36.56 | 35.01 | 33.84 | 32.93 | 32.14 | 31.49 | 30.90 | 30.38 | 29.97 |
| SRCNN [3,4] | 38.01 | 37.21 | 35.87 | 34.40 | 33.28 | 32.30 | 31.94 | 31.85 | 31.04 |
| VDSR [5] | 39.67 | 38.16 | 36.43 | 35.18 | 34.39 | 33.12 | 32.50 | 32.36 | 31.58 |
| Lap-SRN [16,17] | 40.35 | 39.12 | 37.85 | 35.99 | 34.97 | 34.01 | 33.82 | 32.97 | 31.95 |
| Meta-SR [21] | 42.82 | 40.40 | 38.28 | 36.95 | 35.86 | 34.90 | 34.13 | 33.45 | 32.86 |
| RCAN [19] | 42.83 | 40.39 | **38.30** | **36.97** | 35.86 | 34.91 | **34.14** | **33.46** | **32.89** |
| LESRCNN [20] | 42.91 | 40.35 | 38.29 | 36.93 | 35.85 | 34.88 | 34.10 | 33.45 | 32.88 |
| IDMF-SR | **42.83** | **40.40** | 38.29 | 36.95 | **35.87** | **34.92** | **34.14** | **33.46** | 32.88 |

**Table 3.** Average PSNR and SSIM values of different methods under ×2, ×4, and ×8 on datasets Set5, Set14, BSD100, Urban100, and Manga109.

| Method | Scale | Set5 | | Set14 | | B100 | | Urban100 | | Manga109 | |
|---|---|---|---|---|---|---|---|---|---|---|---|
| | | PSNR | SSIM | PSNR | SSIM | PSNR | SSIM | PSNR | SSIM | PSNR | SSIM |
| **Bicubic** | | 33.66 | 0.930 | 30.23 | 0.879 | 29.55 | 0.826 | 26.75 | 0.826 | 30.73 | 0.931 |
| SRCNN [3,4] | | 36.50 | 0.954 | 32.42 | 0.910 | 31.36 | 0.863 | 29.34 | 0.893 | 35.60 | 0.957 |
| VDSR [5] | ×2 | 37.54 | 0.956 | 33.03 | 0.912 | 31.53 | 0.895 | 30.48 | 0.917 | 37.06 | 0.968 |
| Lap-SRN [16,17] | | 37.52 | 0.959 | 33.08 | 0.913 | 31.90 | 0.897 | 30.41 | 0.919 | 37.22 | 0.969 |
| Meta-SR [21] | | 37.10 | 0.957 | 34.18 | 0.911 | 31.88 | 0.910 | 30.52 | 0.932 | 37.35 | **0.985** |
| RCAN [19] | | **38.34** | **0.967** | 34.37 | 0.927 | 32.53 | 0.934 | **33.02** | **0.939** | 39.24 | 0.977 |
| LESRCNN [20] | | 37.65 | 0.9586 | 33.32 | 0.915 | 31.95 | 0.896 | 31.45 | **0.921** | 38.73 | 0.984 |
| IDMF-SR | | 38.20 | **0.967** | **34.38** | **0.930** | **32.57** | **0.938** | 32.96 | 0.920 | **40.15** | 0.980 |
| Bicubic | | 28.30 | 0.810 | 25.98 | 0.639 | 25.79 | 0.668 | 23.04 | 0.658 | 24.86 | 0.787 |
| SRCNN [3,4] | | 30.12 | 0.862 | 26.89 | 0.745 | 26.87 | 0.710 | 24.48 | 0.722 | 27.54 | 0.856 |
| VDSR [5] | ×4 | 31.34 | 0.866 | 27.68 | 0.752 | 27.25 | 0.723 | 25.16 | 0.754 | 28.82 | 0.889 |
| Lap-SRN [16,17] | | 31.45 | 0.885 | 28.17 | 0.769 | 27.32 | 0.736 | 25.21 | 0.756 | 29.17 | 0.890 |
| Meta-SR [21] | | 31.85 | 0.906 | 28.32 | 0.778 | 27.52 | **0.790** | 25.82 | 0.760 | 29.89 | 0.917 |
| RCAN [19] | | **32.62** | **0.912** | 28.89 | 0.790 | **27.99** | 0.751 | 26.88 | 0.812 | 30.97 | **0.921** |
| LESRCNN [20] | | 31.88 | 0.890 | 28.44 | 0.778 | 27.45 | 0.731 | 25.77 | 0.773 | **30.99** | 0.919 |
| IDMF-SR | | **32.62** | 0.910 | **28.90** | **0.792** | **27.99** | **0.790** | **27.10** | **0.818** | 30.98 | **0.921** |
| Bicubic | | 24.40 | 0.656 | 23.06 | 0.567 | 23.67 | 0.545 | 20.74 | 0.516 | 21.48 | 0.650 |
| SRCNN [3,4] | | 25.24 | 0.691 | 23.74 | 0.593 | 24.23 | 0.566 | 21.29 | 0.548 | 22.45 | 0.695 |
| VDSR [5] | ×8 | 25.59 | 0.710 | 24.02 | 0.603 | 24.50 | 0.583 | 21.52 | 0.573 | 23.17 | 0.732 |
| Lap-SRN [16,17] | | 25.92 | 0.728 | 24.28 | 0.614 | 24.54 | 0.590 | 21.67 | 0.582 | 23.40 | 0.759 |
| Meta-SR [21] | | 26.91 | 0.750 | 24.32 | 0.663 | 24.65 | **0.682** | 22.04 | 0.680 | 24.10 | 0.810 |
| RCAN [19] | | 38.34 | 0.795 | 25.43 | 0.668 | **25.16** | 0.614 | 23.50 | 0.653 | 25.47 | 0.826 |
| LESRCNN [20] | | 38.30 | 0.783 | 25.47 | 0.665 | 25.10 | 0.677 | 23.48 | 0.680 | 25.38 | **0.827** |
| IDMF-SR | | **38.35** | **0.796** | **25.50** | **0.669** | 25.10 | 0.674 | **23.51** | **0.682** | **25.49** | 0.827 |

Under ×4, on the Urban100, the PSNR value of IDMF-SR reaches 27.10 dB, which is 1.28 dB and 0.22 dB higher than Meta-SR [21] and RCAN [19]. When the scale factor is 8, on the Set14 dataset, the PSNR value of IDMF-SR reaches 25.50 dB, which is 1.18 dB and 0.07 dB higher than Meta-SR [21], RCAN [19], and LESRCNN [20], as shown in Table 3. It can be seen from the data that when the magnification factor is large and the image details are difficult to recover, the PSNR value of the IDMF-SR is slightly higher than the other algorithms. In summary, from the perspective of objective data, IDMF-SR can effectively restore image details. The objective evaluation index is higher than other algorithms, and the reconstruction effect is good.

### 3.2. Comparison of Subjective Visual Effects

In Figure 9, VDSR [5], Lap-SRN [16,17], Meta-SR [21], and RCAN [19] all optimize details to reduce edge blur. From the overall picture, IDMF-SR and RCAN [19] have similar visual effects to the naked eye. In order to observe the pros and cons of each algorithm more clearly, we select some details of the image to upscale them, and observe the differences in image detail processing of each algorithm, as shown in Figure 9. There is a big difference in the restoration of the detail information of the image. The images (a)–(c) on Set14 img_005 are blurred. Compared with the previous methods, IDMF-SR has an improved reconstruction effect.

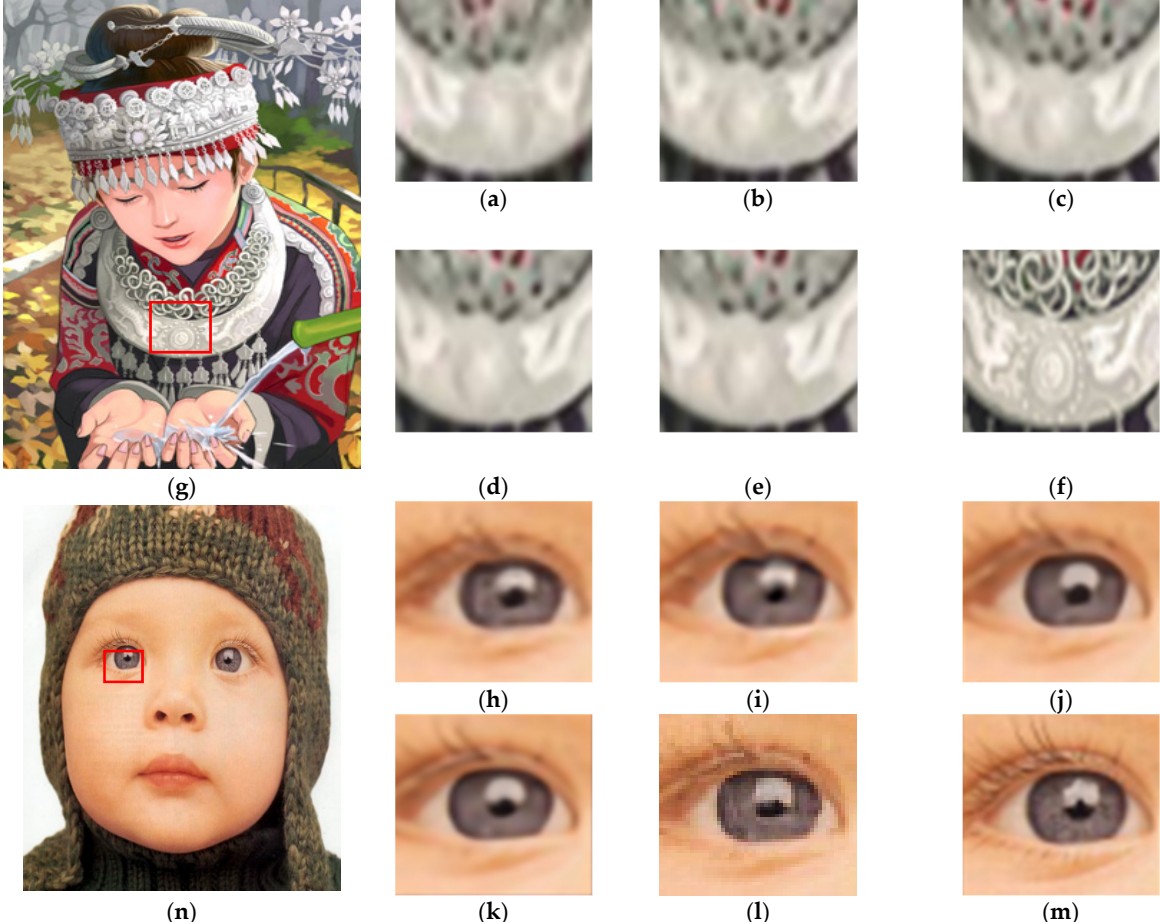

**Figure 9.** The visual effect of each algorithm under ×2 upscale factor. (**a**) VDSR; (**b**) Lap-SRN; (**c**) Meta-SR; (**d**) RCAN; (**e**) IDMF-SR (Ours); (**f**) HR (Original); (**g**) Set14 img_005 (2×); (**h**) VDSR; (**i**) Lap-SRN; (**j**) Meta-SR; (**k**) RCAN; (**l**) IDMF-SR (Ours); (**m**) HR (Original); (**n**) Set5 img_005 (2×).

### 3.3. Comparison of Model Parameters

Compare the traditional algorithms and the IDMF-SR on the Urban100 test dataset Under ×4, the relationship between the average PSNR of the model and the parameter, as shown in Figure 10. The IDMF-SR proposed in this section changes the feature learning module based on Meta-SR [21], adopts an information distillation structure, progressively extracts image features, and cascades features. The feature does not fully participate in the next stage of the feature learning task. Therefore, only a few parameters can be used to achieve fast and accurate image super-resolution reconstruction, preventing parameter redundancy. It can be seen from Figure 10 that IDMF-SR has a 69.8% reduction in parameter quantity than Meta-SR [21] and a 2% increase in PSNR value. The algorithm in this section makes a trade-off between the number of model parameters and the PSNR value, which

not only ensures the improvement of the SISR performance but also reduces the number of parameters.

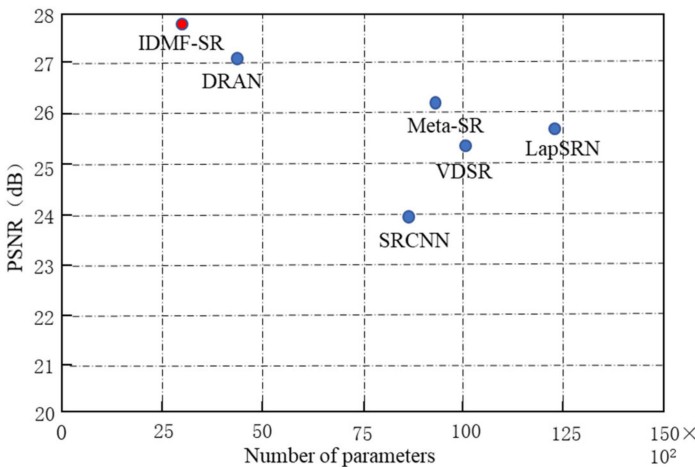

**Figure 10.** Multiple model PSNR value and parameter quantity relation diagram.

## 4. Discussion

*Ablation Studies of IDM and CCAM*

To quickly demonstrate the effect of the information distillation module (IDM) and contrast-based channel attention mechanism (CCAM), we remove the IDM between IDB and/or CCAM, so the IDMF-SR becomes the basis of a deep network, which we named IMDN-Basic, as described in Figure 11. Firstly, we use four IDB to certify the effect of IDM and CCAM. In Table 4, when both IDM and CCAM are removed, the PSNR on Set5 at the scale factor of 4 is 32.48 dB as the first column. When CCAM is added, the PSNR value reached 32.56 dB. This is because CCAM can improve the information about structures, textures, and edges that are propitious to enhance image details. The PSNR value reaches 32.62 dB with the contribution of IDM and CCAM. This indicates that IDM and CCAM are essential for improving SISR performance.

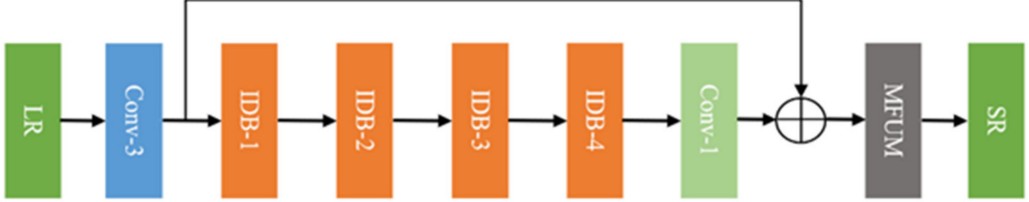

**Figure 11.** IMDN-Basic. The grey box represents multi-factor upscaling module.

**Table 4.** Investigations of CCA module and IIC scheme.

| Different Combination of IDM and CCAM | | | |
|---|---|---|---|
| IDM | ✗ | ✗ | ✓ | ✓ |
| CCAM | ✗ | ✓ | ✗ | ✓ |
| PSNR on Set5 (×4) | 32.48 | 32.56 | 32.60 | 32.62 |

## 5. Conclusions

In this paper, we propose an information distillation structure to progressively extract multi-scale spatial features to achieve fast and accurate image super-resolution. The information distillation module divides the captured feature map into two parts. After each level of convolution, one third of the feature maps are retained and cascaded after the last convolutional layer. CCAM can further enhance image details, such as edges, textures, and

structures. In addition, we propose a multi-factor upsampling module, which uses scale factors to predict filter weights. IDMF-SR can train a single model for super-resolution of arbitrary scale factor to achieve image super-resolution. Extensive experiments illustrate that the proposed IDMF-SR outperforms state-of-the-art versus SISR in terms of qualitative and quantitative evaluation.

**Author Contributions:** Project administration, S.C.; Validation, Z.L. and Y.C.; Visualization, N.Z. and Y.C.; Writing—original draft, Y.C.; Writing—review & editing, Y.C. and S.C. All authors have read and agreed to the published version of the manuscript.

**Funding:** This research received no external funding.

**Institutional Review Board Statement:** Not applicable.

**Informed Consent Statement:** Not applicable.

**Data Availability Statement:** Not applicable.

**Conflicts of Interest:** The authors declare no conflict of interest.

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
