# Peer review of "Multi-Scale Factor Image Super-Resolution Algorithm with Information Distillation Network"

_applsci, doi:10.3390/app12094131_

Round 1

Reviewer 1 Report

The manuscript entitled “Multi-scale Factor Image Super-Resolution Algorithm with Information Distillation Network” presents a study of the use of MFN to generate super-resolution images with low memory consumption. The article is well-structured and presents an interesting topic. However, before considering it suitable for publication in the Applied Sciences journal, the authors must review the document and clarify some topics.

First, the entire text needs an English grammar review. The document has excessive use of the first plural person, “we.” From abstract to the conclusion. Also, the authors must improve the case-study description.

For a more detailed review, please follow the comments by section:

Introduction

The introduction section provides a background about the study, focusing on the gap that the work will fill. It can be considered the state-of-art but should be improved with more articles citations to corroborate the gap. In the final section, the authors present the contribution of the paper. This list should be formatted as specific objectives instead of activities.

Minor comments:

  • Abbreviations should be next to the names: Single Image Super-Resolution (SISR). They should be explained in the first moment it appears.
  • Figure 1 should be after its citation in the text (line 76, page 2).
  • Excessive use of Authors' names in the citations. As the citation style is by number, it should avoid the excessive use of “Authors Name et al. [##]. “

Materials and Methods

This section explains the methodology well, especially the steps on the network structure provided in sub-section 2.1. My only concern is about the “materials,” which is the title of this section, and it is not shown. In this case, sections 3.1 (Datasets and Evaluation Metrics) and 3.2 (implementation Details) should be sub-sections here.

Results

Expecting that the authors change the 3.1 and 3.2 subsections to Materials and Methods, the authors should improve the explanation about the Comparison with State-of-Art Algorithms (subsection 3.3). It can be seen the results, but it is not explained how the comparison is made.

Discussion

Please, improve the description of Figure 11 in the main text and the figure caption.

Conclusion

The conclusion needs to have more robust comments. It is too shallow, saying that the results are promising. In this section, the authors must show that the objectives and the proposals done in the introduction are complete.

Reviewer 2 Report

The paper presents a deep learning method for image super-resolution.
The method is composed of two steps: an information distillation module and a multi-factor upsampling module.
The method was evaluated on several datasets from literature.
The method outperformed several state-of-the-art methods on the chosen datasets.

The writing is problematic, with unusual syntax and dangling subordinates.
The abstract is especially badly written.

The novelty of the proposed method is limited, as there are other papers which propose information distillation for image super-resolution.

Please define "HR", "LR" and "SR" somewhere in the paper. I assume: high resolution, low resolution and super-resolution?

At line 120: "What I need to stress here is that the features are actually the same but sent to different parts of the network." This part is unclear. Which features are the same? Also, the first person is unusual.

In the method, I could hardly understand anything from line 170 onward.
Firstly, you introduce a distinction between integer and non-integer scaling factor, but then you (correctly) state that it is irrelevant: you can always find a pixel in LR which corresponds to a pixel in SR using nearest-neighbor downsampling of SR. So why is the distinction stated in the first place? Could you just remove the whole paragraph?
Also, "these two-point pixels are called the most relevant pixel pair." You just stated that there is a single pixel. What is a two-point pixel? This should be rewritten and clarified.
Sub-subsections (2) and (3) should also be improved. If I understand correctly, the network input includes the offset in equation 14 somehow. However, both a high-level network overview and the network details are missing.
At line 199: "and then predicted filter weight" please clarify.

Round 2

Reviewer 1 Report

The new version of the manuscript has improved with reviewers' recommendations.